# Can Creativity and Cognitive Reserve Predict Psychological Well-Being in Older Adults? The Role of Divergent Thinking in Healthy Aging

**DOI:** 10.3390/healthcare12030303

**Published:** 2024-01-24

**Authors:** Giulia Fusi, Jessica Giannì, Virginia Maria Borsa, Laura Colautti, Maura Crepaldi, Massimiliano Palmiero, Francesca Garau, Salvatore Natale Bonfiglio, Ylenia Cao, Alessandro Antonietti, Maria Pietronilla Penna, Luca Rozzini, Maria Luisa Rusconi

**Affiliations:** 1Department of Human and Social Sciences, University of Bergamo, 24129 Bergamo, Italy; jessica.gianni@unibg.it (J.G.); virginiamaria.borsa@unibg.it (V.M.B.); maura.crepaldi@guest.unibg.it (M.C.); marialuisa.rusconi@unibg.it (M.L.R.); 2Department of Psychology, Catholic University of the Sacred Heart, 20123 Milan, Italy; laura.colautti1@unicatt.it (L.C.); alessandro.antonietti@unicatt.it (A.A.); 3Department of Communication Sciences, University of Teramo, 24100 Teramo, Italy; mpalmiero@unite.it; 4Department of Pedagogy, Psychology, Philosophy, University of Cagliari, 09123 Cagliari, Italy; fgarau.91@gmail.com (F.G.); nbonfiglio@fatebenefratelli.eu (S.N.B.); ylenia.cao95@gmail.com (Y.C.); penna@unica.it (M.P.P.); 5Department of Clinical and Experimental Sciences, University of Brescia, 25136 Brescia, Italy; luca.rozzini@unibs.it

**Keywords:** aging, cognitive reserve, divergent thinking, well-being, emotional competence, health

## Abstract

The maintenance of psychological well-being (PWB) in the older adult population is a pivotal goal for our rapidly aging society. PWB is a multicomponent construct that can be influenced by several factors in the lifespan. The beneficial role of divergent thinking (DT) and cognitive reserve (CR) in sustaining older subjects’ PWB has been scarcely investigated so far. The present study aims to investigate the relationships between DT, CR, and PWB in a sample of 121 healthy older adults (61 females; M age: 73.39 ± 6.66 years; M education: 11.33 ± 4.81 years). The results highlight that better DT performance predicts higher CR, which mediates an indirect positive effect of DT on emotional competence, one of the PWB factors. It follows that DT and CR can be considered protective factors in aging, and their effects go beyond cognitive functioning, revealing a positive effect even on some PWB components. The practical implications regarding targeted health interventions for prevention in the older adult population to support well-being and promote healthy aging are discussed.

## 1. Introduction

### 1.1. Psychological Well-Being in Aging

Life expectancy has consistently grown over the years [1], and the number of older adults continues to grow worldwide [2]. Research has revealed that psychological well-being (PWB) plays a key role in older adults’ health, both reducing the risk of developing chronic diseases (lung diseases, cognitive decline, cardiovascular diseases, etc.) and promoting longevity [3,4,5,6,7]. Therefore, international organizations have stressed the need to promote mental health and psychological well-being in all stages of life [8,9,10].

A few decades ago, Carol Ryff [11] defined PWB as a multidimensional construct based on the eudaimonic component, which concerns personal development and growth, as the realization of individual potential [12]. Conversely, the hedonic dimension conceives PWB as the personal pleasure linked to positive sensations and emotions [13].

Aging—which is a complex phenomenon characterized by a variety of changes, including physical, emotional, and cognitive ones [14]—has been generally associated with a decline in all these domains. However, aging has also been associated with improved levels of PWB, mainly in the emotional sphere. According to the so-called “well-being paradox of aging”, older adults experience high levels of affective well-being [15] and increasing emotional stability [16]. According to the “positivity effect”, older adults tend to focus on emotional well-being when the time perspective becomes more limited [16] and tend to be oriented to positive emotional stimuli rather than to negative ones [17,18]. Moreover, according to the “Strength and Vulnerability Integration Model” (SAVI) [19], older adults perceive higher well-being because of their accumulated life experience, which allows them to develop powerful cognitive and behavioral control strategies. In particular, it has been hypothesized that an emotion–cognition interaction mechanism, which involves both emotional processing and emotional regulation, together with cognitive control mechanisms and strategies, can, at least partially, explain these effects and be linked to long-term outcomes, such as real-world behavior and perceived PWB [16,20].

However, the evolution of the different aspects of PWB, especially in the eudaimonic dimension, throughout the lifespan is more complex than previously thought [21], since the concept of PWB conceives multiple aspects that go beyond the emotional sphere. Some aspects of PWB tend to decrease during the last decades of life, e.g., personal growth and purpose in life [11,22], while others generally remain stable (environmental mastery) or even improve (e.g., positive relations [11,23]). The trajectories are therefore complex, often nonlinear, and can be influenced by several cognitive and social factors [24]. The identification of these factors is therefore pivotal in the context of successful aging, especially focusing on the eudemonic approach to PWB and its diverse components (e.g., coping strategies, emotional competence, and personal satisfaction [1,25]).

### 1.2. Correlated Variables

A pervasive individual heterogeneity in the older population is evident, and it holds in several domains: engagement with life, risk of diseases, cognitive and mental health, physical activity, and PWB [26]. According to this notion, several studies have suggested that different factors, such as genetic, physiological, metabolic, and environmental effects, dietary habits, lifestyle, social relationships, and some positive psychological characteristics, could contribute to successful aging and might have an impact on the older adult’s perception of PWB [24,27,28,29,30,31]. This is in line with the positive aging literature, which considers optimal, successful, productive, and healthy aging as the result of the interplay of five independent factors: health, cognition, activity, affect, and physical fitness [32]. In addition to these variables, individual differences in cognitive abilities [17,22,33], such as divergent thinking (DT) [34] and cognitive reserve (CR) [35], have started to attract attention from researchers, especially because of the previously mentioned emotion–cognition interaction in the perception of PWB.

#### 1.2.1. Cognitive Reserve

Among these factors, brain maintenance [36] and compensation abilities have been identified as crucial mechanisms that account for interindividual variability in cognitive aging [37]. In particular, CR has been introduced to explain this phenomenon [38]. As Stern explained [39,40], higher levels of CR involve the recruitment of different brain networks that can facilitate the production of alternative strategies, thus leading to an alternative approach to problems to better cope with cognitive decline and a positive influence on well-being. A lot of proxies have been used as indicators of CR. Among these, it is usually assumed that cognitive reserve is built up during the lifespan, considering the individual’s years of education, type and complexity of working activity, and leisure-time activities [41]. Consistently, some studies reported that older adults’ engagement in more diverse activities is associated with higher PWB over time [42]. Other research highlighted the helpful role of social integration in compensating for losses typically experienced in later adulthood [43]. CR has been long investigated, and its protective role in global cognitive functioning has been proven by several studies [38,39,40,44,45], as it has been associated with increased strategy selection abilities, memory, speed processing, and, in general, greater executive functioning [46].

Efficient cognition, especially the development and use of cognitive control strategies, has been associated with adaptive psychological processes, such as goal-pursuit behaviors [47], less usage of avoidant coping strategies, greater management of age-related stressful events, and emotion regulation abilities, thanks to the implementation of self-regulation behaviors [17,35].

However, there are very few studies examining the direct relationship between CR and well-being in the older population. According to some authors, higher CR is associated with a higher quality of life in older adults [48], as well as cognitive well-being [49], while a greater educational level and participation in cognitively stimulating leisure activities (two proxies of CR) showed negative correlations with depressive symptoms in later life [50,51]. Due to the variety and heterogeneity of these findings, further clarifications might be crucial to shed light on such relations, especially on their impact on perceived PWB in older adults.

#### 1.2.2. Divergent Thinking and Its Relationship with Cognitive Reserve

On the other hand, a cognitive skill that could be preserved in aging, and that is strictly related to control and executive mechanisms, is divergent thinking (DT; [52]). This concept was first introduced by Guilford [53,54], who defined DT as the ability to think freely in multiple directions, allowing one to find a variety of potentially feasible solutions to open-ended problems. It is described by four indicators: fluency (number of ideas), flexibility (ability to shift into different categories), originality (number of unusual ideas), and elaboration (number of details that enrich an idea). DT has been compared to the cognitive process that leads to original-idea production [55], as well as an indicator of creative potential and a predictor of creative achievement [56].

The trend of DT abilities in the elderly population is still controversial, especially because of the complexity and multidimensionality of the construct [57]. However, recent evidence [58] showed that DT does not decline steadily during the last decades of life. DT abilities can be, at least partially, spared under specific conditions: if no time constraints are imposed, if the workload is not too challenging [57], and if the type of employed material and the involvement of different underlying cognitive processes (e.g., verbal or visuospatial) are taken into account [57,59].

Considering that DT abilities can be spared in the aging population and that healthy aging usually means adapting behaviors to the new demands of the environment, some authors have proposed that one way to address these challenges might depend, at least partially, on older adults’ DT skills, which are therefore potentially significant for the perception of PWB [34]. A previous study used a structural equation modeling approach to prove that older adults with higher scores in DT have better PWB [34]. However, a differential impact of DT on the diverse components of PWB was not considered in the model. Other studies highlighted that creative thinking, encouraging the development of new and existing competence, may promote and valorize psychological resources, such as self-efficacy and coping strategies, as well as cognitive processes [60,61]. However, since only a few studies investigated the relationship between DT abilities and PWB in the older adult population, further clarification is still needed, especially on the impact of DT on the different aspects of PWB.

It is worth highlighting that DT and creative cognition are also described as the ability to access and apply alternative strategies, to keep an open mind, to establish new and unusual relationships, and to change perspective [62]. Such abilities are similar to those hypothesized for CR, which potentially allows subjects to use alternative strategies to better cope with age-related brain changes [63]. Assuming that DT may induce older people to use alternative paradigms and strategies when standard approaches become unavailable to face difficulties that occur in daily life [64], some studies have also conceptualized it as an indicator or proxy of CR in the elderly [63,64]. The existing literature has confirmed this hypothesis and found a positive relation between DT and CR [63,64]. Mainly the verbal (rather than visual) components of DT seem to positively predict subjects’ CR [64].

However, the relationships between DT, CR, and PWB over time, as well as whether and how they impact the perception of PWB in its diverse facets in older adults, require further clarification.

### 1.3. Aims and Hypotheses

The main aim of the present study was to shed light on the relationships between DT, CR, and PWB in the aging population. In particular, it aimed to shed light on the relationships between (1) verbal and figural DT and CR, (2) CR and PWB, considering the eudaimonic perspective, and (3) DT and PWB. According to the literature, we hypothesized that DT would significantly and positively predict CR [63,64], CR would be positively associated with PWB, and a positive relationship between DT and PWB would also emerge.

## 2. Materials and Methods

### 2.1. Participants, Study Design, and Procedure

A total of 135 healthy older Italian adults were randomly recruited from two Italian regions, Lombardy (*N* = 92) and Sardinia (*N* = 43). All participants underwent an assessment to be screened for the inclusion criteria: age ≥60 years old (to be considered older adults according to the World Health Organization, https://www.who.int/health-topics/ageing#tab=tab_1 (accessed on 15 January 2021); Mini-Mental State Examination (MMSE) ≥24 [65]; and no history of vascular, neurological, or psychiatric disorders.

After the check for inclusion criteria, two subjects were removed due to an MMSE under the selected cut-off. Different exclusion criteria were used according to the aim of the present work: 12 subjects were removed due to the incompleteness of data collected on creative abilities. The final sample is therefore composed of 121 older adults (61 females and 60 males; mean age: 73.39 ± 6.66 years, range: 63–88 years; mean educational level: 11.33 ± 4.81 years, range: 3–27 years).

Data collection was performed in person from April 2021 to December 2022; this sample was already described by Colautti and colleagues (2023) [59]. All participants read and provided informed consent. Two individual in-person sessions with an experienced researcher were scheduled in a quiet setting, and each of them lasted approximately 60 min. All participants took part in the study on a voluntary basis, and no incentive was provided.

The present research used a correlational design, and post hoc power analyses conducted with G*POWER (version 3.1.9.3) calculated with alpha = 0.05, effect size calculated using the mediation R2, i.e., 0.13, sample size = 120 and 3 predictors) evidenced that the study was adequately powered (power = 0.987).

The research protocol and procedure were approved by the institutional ethical committees of the University of Bergamo and of the Catholic University of the Sacred Heart in Milan. The study was conducted following the Declaration of Helsinki, and all the participants signed informed consent forms.

### 2.2. Materials

#### 2.2.1. Psychological Well-Being

BEN-SSC (BEN-essere e Invecchiamento) [25] is a 37-item questionnaire designed for the assessment of psychological well-being specifically in the older adult population. The questionnaire is grounded on the concept of psychological well-being proposed by Carol Ryff [66], who considers well-being to be a multidimensional construct. In addition to the original domain of the Ryff model, BEN-SSC also includes the concept of self-efficacy (see [25]), which is considered by the authors to be of particular importance in the study of aging. In particular, after Italian validation and a factorial analysis, BEN-SSC considers the following subscales: personal satisfaction (BEN_PS), coping strategies (BEN_CS), and emotion regulation skills (BEN_EC, i.e., emotional competence). Subjects are asked to respond using a 4-point Likert scale (ranging between 1 = not at all and 4 = yes/often). All items are self-descriptive and formulated to assess positive attitudes and beliefs rather than negative ones. The total score (BEN_tot; maximum = 148) is obtained by summing all items’ responses; the higher the score is, the greater the level of psychological well-being is. The reliability of the questionnaires was very good (Cronbach’s a = 0.91).

#### 2.2.2. Cognitive Reserve

The Cognitive Reserve Index questionnaire (CRIq, [41]) is a 20-item questionnaire designed to evaluate the amount of CR acquired during a person’s lifetime. It is composed of three sections that correspond to the sources of CR: the individual’s education (CRIq school; CRIq_S), occupation experiences (CRIq work; CRIq_W), and activities carried out during leisure time (CRIq leisure activities; CRIq_LA). The items aim to investigate the type and frequency of activities carried out from the age of 18 up to the present. In particular, they consider the individual’s engagement in weekly, monthly, and annual activities. Three different scores are provided to quantify cognitive reserve related, respectively, to school, work, and leisure activities. These three subscores are summed to obtain the total score; scores < 70 correspond to very low levels of CR, those >130 correspond to very high levels, and those between 85 and 129 correspond to medium levels of CR.

#### 2.2.3. Divergent Thinking

Two tasks selected from the TTCT (Torrance Tests of Creative Thinking [67,68]) were assessed to evaluate both verbal and visual DT.

The Parallel Lines Test (LT) assesses figural DT. Participants are asked to complete as many figures as possible within 10 min by drawing from two parallel lines, with a total of 30 pairs of parallel lines provided. Items are scored according to the manual and considering (i) the number of reliable answers given, (ii) the number of semantic categories in each answer, and (iii) the number of innovative responses, representing a measure of fluency, flexibility, and originality, respectively. In particular, the flexibility value tells us about the variety of shifts in responses. Originality is scored according to the manual’s sample (0 points = responses provided by ≥5% of 500 people; 1 point = responses provided by 2–4.99% of 500 people; 2 points = responses provided by <2% of 500 people or responses not listed in the manual). A figural creativity (Lines_M) index was developed by averaging the standardized scores of the above-mentioned subscales (i.e., fluency, flexibility, and originality).

The Alternative Uses Task (AUT) assesses verbal DT. Participants are asked to find as many different and original cardboard box uses as possible within 10 min. Items are scored according to the manual’s criteria, considering the above-mentioned indexes: fluency, flexibility, and originality. A verbal creativity (AUT_M) index was developed by averaging the standardized scores of these subscales.

Moreover, according to previous studies (see, e.g., [64]), a general DT index (DT_tot; divergent thinking composite) was also computed by averaging the standardized scores of all subscales of the verbal and figural tasks.

## 3. Data Analyses

Jamovi version 2.3.21.0 for Mac was employed for all data analyses. Before running parametric analyses, we checked for the normal distribution of the data using criteria suggested by West et al. [69] and by Kline [70] (±2 for asymmetry and ±7 for kurtosis). All variables were normally distributed according to these criteria.

We exploratively performed *t*-tests to control for gender differences and correlations to investigate the relationships between age and the DT, CR, and PWB subscales (i.e., DT, CR, and PWB). Then, we computed Pearson’s correlation coefficients to explore the relationships between the previously mentioned variables, including significant results in subsequent hierarchical and GLM mediation models.

## 4. Results

The descriptive statistics of the assessed measures are reported in Table 1. Moreover, considering that all variables complied with the assumption of homogeneity of variances (non-significant Levene tests), several independent-sample *t*-tests were computed to check for gender differences (see Table 1).

Only the CRI subscores related to work and school were significantly different between males and females, with males showing higher scores in both variables (CRIq_school: males M = 111.92, SD = 14.23; female: M = 106.11, SD = 14.92; CRIq_work: males M = 108.33, SD = 19.37; female M = 100.39, SD = 21.28).

The correlations between age and the other variables are reported in Table 2.

Age correlated negatively with many of the considered variables: with all the figural DT indexes, with verbal originality and total score, with all CR indexes, and with emotional competence and coping strategies from well-being subscales.

The Pearson’s correlation coefficients are shown in Table 3.

### 4.1. Divergent Thinking and Cognitive Reserve

Consistent with our hypothesis, we found a significant positive and moderate correlation between the DT composite score and the total index of CR (r = 0.34, *p* < 0.001). All the figural DT indexes showed positive and moderate (r from 0.23 to 0.40) correlations with the total CR index. However, only the fluency verbal index correlated with the total CR (r = 0.23, *p* = 0.013).

Observing the relationships between the subscales, moreover, it can be seen that all of the DT indexes (i.e., fluency, flexibility, and originality) correlated (r from 0.20 to 0.35) with all of the CR subscales (i.e., school, work, and leisure activities), except for the originality index, which does not correlate with the school-related CR index. In contrast, with regard to verbal DT, the indexes correlate only with leisure-related CR (r from 0.20 to 0.28).

### 4.2. Cognitive Reserve and Well-Being

A significant and positive correlation was observed between the total PWB index and the total CR index (r = 0.20, *p* = 0.025; see Table 3). In particular, CR (except the school subscale) correlates primarily with the older adults’ perceived emotional competence (BEN_EC) (r from 0.28 to 0.34). Positive correlations were also observed between CRIq work and the coping strategies subscale (BEN_CS).

Following these results, we ran four hierarchical multiple regression analyses considering the three CRIq subscales as predictors and BEN total PS, EC, and CS as dependent variables (DVs), controlling for the age effect. The regression model with BEN_SP as the DV was not significant. On the contrary, the other three models proved to be significant and can be observed in Table 4.

### 4.3. Divergent Thinking and Well-Being

Contrary to our expectations, only one marginal correlation was observed between the originality index of figural DT and PWB related to emotional competence (r = 0.18, *p* = 0.046). No other statistically significant correlations between DT skills and PWB were found.

However, considering the results described in the previous paragraphs (i.e., DT was positively correlated with CR, and CR was, in turn, positively correlated with well-being), we tested whether the relationship between DT and PWB might be indirect and mediated by CR. We, therefore, conducted a General Linear Model mediation analysis, controlling for the age effect, with age and DT as predictors, CR as a mediator, and PWB, in the emotional competence component, as the dependent variable (see Figure 1 for path diagram visualization).

The parameters and the results of the mediation analysis are shown in Table 5.

We found that BEN_EC was directly predicted by neither age nor DT_tot. However, it was positively predicted by the total CR score; both predictors are significantly related to CRI_tot. Moreover, interestingly, both indirect effects of age and DT, through the mediation of CRIq_tot on BEN_EC, were significant.

## 5. Discussion

The present study aimed to investigate the relationships between DT, CR, and the different facets of PWB. In particular, the study considers the multidimensional construct of PWB through the eudaimonic perspective [11]. A sample of 121 healthy older Italian adults was enrolled.

### 5.1. The Effect of Age Progression

Although age did not correlate negatively with the total score of the perceived PWB of the older adults in our sample, we found that age had slight significant negative correlations with two subscales: coping strategies and emotional competence. So, even though all subjects generally perceived stable psychological well-being and maintained good personal satisfaction, they perceived, on the contrary, a slight decline in their coping and emotional competence during the last decades of life.

These results are, at least partially, in line with previous studies pointing out that the relationship between age and the multidimensional construct of well-being is more complex than previously believed, e.g., [21]. Some components of PWB remain more stable or even increase, e.g., [11,23], while others tend to decrease, e.g., see [11,22].

### 5.2. Divergent Thinking and Cognitive Reserve: What Is New?

The first research question regarded the relationship between DT abilities and CR. In line with previous research [63,64], which found a positive prediction of the CR index by DT abilities, our data evidenced a positive and moderate correlation between the total score of divergent thinking abilities and the total CR index. The higher the creative potential of older subjects is, the higher their cognitive reserve appears. This confirms the hypothesis that higher creative potential and capacity might have a positive influence on CR and therefore on its moderating effect on cognitive efficiency and functioning [38,39,43,44,63,71,72]. However, going deeper into the data, unlike previous studies, we found that CR had a stronger relationship with figural DT thinking, rather than with verbal DT. Indeed, all the indexes of figural DT (i.e., fluency, flexibility, and originality) correlated significantly and positively with all CR indexes (i.e., school, work, and leisure activities), while verbal DT seems to be positively correlated only with the subscores of leisure activities. This sheds light on potential new mechanisms, evidencing that not only verbal but also figural divergent thinking skills can have a positive effect on cognition and be targeted by stimulation programs, even though these abilities are more sensitive to age decline.

Other studies are needed to confirm these and previous results to generalize the evidence about the influencing role of verbal vs. figural DT on CR.

### 5.3. Cognitive Reserve and Well-Being

Previous research has already found that CR mitigates the effect of age on cognitive functioning [38,43,73,74]. Furthermore, it can also help to protect the aging population from the effects of stress [75,76], is linked to a higher perceived quality of life [48], and influences well-being. However, only a few studies have directly and thoroughly investigated the link between CR and the PWB perceived by older adults in all its diverse facets. The present study evidence that, according to the hypothesis, higher CR predicts higher levels of PWB. However, the link between CR indexes seems to change according to the PWB subcomponent considered. Perceived coping strategy abilities and emotional competence were predicted by specific indexes of CR, respectively, by work and by both work and leisure-time indexes. Thus, older adults who have accumulated more years of work or held jobs requiring more cognitive and behavioral flexibility seem to perceive both better coping strategies and higher emotional competence, which means a higher perception of understanding others’ emotions, being more satisfied with their social relationships, being more intuitive and confident in sharing their experiences, and being confident in asserting their opinions with relatives and friends [1,25]. The implications of these findings will be discussed in the next paragraph.

### 5.4. Divergent Thinking and Well-Being: The Mediating Role of Cognitive Reserve

Contrary to our expectations, we did not find a direct relationship between DT and PWB. Only a marginal positive correlation was found between the originality of figural DT responses and the subscale of emotional competence. However, considering the previously presented results and the strong relationship between the components of the variables of interest and the emotional competence subscale, we performed a mediation analysis. It showed a significant indirect effect of DT on the perceived emotional competence of older adults through the mediation of the CR index, also controlling for the (negative) age effect. It seems that, in the aging population, having higher DT predicts higher scores of cognitive reserve, which, in turn, predicts higher levels of PWB, but only in its emotional competence subscore.

Considering these results in the context of the link between the cognitive and emotional mechanisms [16,19,20] that can underlie both positive and negative effects on PWB in older subjects, two main findings can be highlighted. On the one hand, in line with previous findings, DT can be considered a resource and a protective factor during the aging process [63,64,77]. Subjects with higher DT tend to accumulate higher CR, which, according to a wide range of evidence, can support cognitive efficiency during both healthy [44,78] and pathological aging [79,80,81]. On the other hand, the beneficial effect of CR seems to go beyond cognitive efficiency, significantly and positively predicting some PWB facets (mainly coping and emotional competence), confirming a strict interplay between cognition and emotions during the aging process, e.g., [16,20]. Additionally, our results evidence that having higher DT potential, as well as predicting CR, exerts a positive effect (through CR) on the emotional competence component of well-being, suggesting that people with higher DT tend to be involved in more complex jobs and to have heterogeneous interests (both personal and social) in their free time during life; this, in turn, positively influences their well-being, since they feel more competent in the area of emotional skills.

Thus, although, according to the literature, older adults maintain a generally good level of PWB (i.e., “well-being paradox of aging” [15]), the data in this study suggest that the aging process can instead still harm the perception of being competent in the emotional domain. However, our data also show that the negative effect can somehow be mitigated by cognitive factors such as DT and CR. These findings might shed light on new mechanisms of interaction between cognitive and emotional aging: in particular, one explanation might be found in the evidence that—according to the SAVI hypothesis [19]—older adults develop powerful cognitive and behavioral control strategies through their accumulated life experience. Control mechanisms can thus be considered one of the main links in cognitive–emotional interaction, and their role also seems to be crucial in the perception of PWB [17]. Both DT and CR, indeed, are highly related to executive functioning [52,82,83,84,85,86,87,88], to better fronto-parietal functioning and connectivity [89,90], and, therefore, to control mechanisms. Consequently, it is hypothesized that higher DT abilities, which implies higher executive functioning, might enhance the connectivity in the fronto-parietal control network, expanding CR and the flexibility of control mechanisms, which, in turn, allow older subjects to perceive higher competence not only in the cognitive but also in the emotional domain. These results, taken together, confirm that PWB is a non-unitary construct since its components can be influenced by aging processes and by cognitive, emotional, and social variables in different ways, making the trajectories nonlinear and complex.

To conclude, cognitive and mental health, and therefore good PWB, in the aging population are important objectives for both economic and health policy, considering their protective effects on several age-related conditions. Future studies that integrate the findings on variables that can impact older adults’ well-being while also considering the role of DT and CR might be pivotal for a comprehensive, multidisciplinary, and multi-level comprehension and the promotion of PWB in this population.

### 5.5. Strengths and Limitations

The present study has some strengths. First, the enrollment of an adequately large sample of Italian older adults has allowed the achievement of good statistical power. Second, as reported in the previous section, the findings allow us to take a step forward in the understanding of the intervening variables in the perception of PWB and healthy aging, with consequent practical implications for the planning of interventions for older adults’ cognitive and mental health.

However, this study might suffer from some limitations. First, the older adults enrolled in the present study came from only two areas of Italy, and this may have, in part, influenced the results, limiting the generalizability of the results. Second, we used only one DT task for both the verbal and figural domains. However, this choice was made given the age of the subjects (i.e., considering fatigue). Further studies might consider using multiple tasks for each domain to confirm these results and even better understand the implications of the cognitive–emotional control mechanisms involved in later adulthood. Lastly, the study has a cross-sectional and correlational design: longitudinal studies might be needed to confirm the prediction of our regression and mediation models.

## 6. Conclusions

This research shows that some PWB components, such as coping strategies and emotional competence, might be negatively affected by age, but some variables can positively influence this relationship. The results indeed support the evidence of an indirect positive effect of DT on PWB through the mediation of CR, particularly on the older adult’s perception of having good emotional competence. These findings confirm the pivotal roles of DT and an active life and lifestyle (i.e., mainly work and leisure activities) in building greater CR, helping older adults not only to maintain efficient cognitive functioning but also to perceive higher PWB. These findings can have important practical implications for health prevention and intervention for older adults, unraveling innovative ways to enhance this population’s cognitive and mental health: DT might thus be considered a novel target of intervention. This would help older adults find and use different solutions and strategies to face the challenges of daily life, making them feel more confident and capable in their cognitive and emotional skills and therefore more engaged in activities. These, in turn, would increase adults’ and older adults’ CR, protecting them from cognitive decline but also having a positive influence on their PWB.

## 7. Implications and Future Directions

Future studies could better untangle the possible influence of CR and DT on other aspects of well-being as well (e.g., hedonic well-being), given its complex and non-unitary nature. Moreover, it could be useful to analyze whether such an influence can also be a protective factor for other cognitive and emotional skills related to emotional competence (e.g., perception or recognition of one’s own or others’ emotions) and not only for self-reported perception. All this evidence would have significant practical implications for targeted health interventions, both for prevention and for training programs in the elderly population, to sustain both their cognitive and emotional skills.

## Figures and Tables

**Figure 1 healthcare-12-00303-f001:**
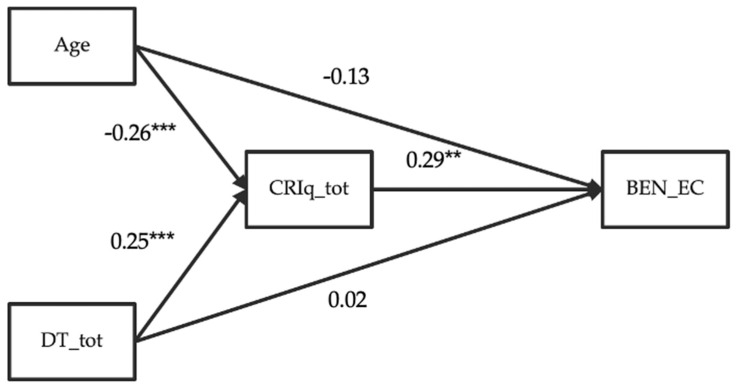
Visual representation of the GLM mediation analysis investigating the role of CR in mediating the relationship between DT skills and the subscale emotional competence (PWB), controlling for the effect of age. ** *p* < 0.01; *** *p* < 0.001.

**Table 1 healthcare-12-00303-t001:** Descriptive statistics (mean and standard deviations) for divergent thinking, cognitive reserve, and psychological well-being variables.

		Mean	SD	t	*p*	d
BEN_tot	sample	112.62	15.10	0.90	0.368	-
	male	111.37	15.96			
	female	113.85	14.23			
BEN_PS	sample	33.73	5.85	1.08	0.284	-
	male	33.15	5.75			
	female	34.30	5.95			
BEN_EC	sample	30.88	5.04	1.40	0.165	-
	male	30.23	5.45			
	female	31.51	4.55			
BEN_CS	sample	26.39	4.38	−0.65	0.517	-
	male	26.65	4.40			
	female	26.13	4.38			
CRIq_tot	sample	114.82	20.52	−1.10	0.274	-
	male	116.88	20.66			
	female	112.79	20.33			
CRIq_S	sample	108.99	14.81	−2.19	**0.031 ***	−0.40
	male	111.92	14.23			
	female	106.11	14.92			
CRIq_W	sample	104.33	20.66	−2.15	**0.034 ***	−0.39
	male	108.33	19.37			
	female	100.39	21.28			
CRIq_LA	sample	120.53	24.69	0.80	0.425	-
	male	118.72	24.90			
	female	122.31	24.55			
Lines_Flu	sample	9.72	5.66	−1.25	0.213	-
	male	10.37	5.59			
	female	9.08	5.70			
Lines_Flex	sample	7.17	3.95	−0.97	0.334	-
	male	7.52	3.90			
	female	6.82	3.99			
Lines_Orig	sample	12.63	8.98	−0.67	0.502	-
	male	13.18	9.05			
	female	12.08	8.96			
AUT_Flu	sample	6.98	3.61	−0.45	0.651	-
	male	7.13	3.75			
	female	6.83	3.50			
AUT_Flex	sample	5.05	2.56	−0.92	0.357	-
	male	5.27	2.85			
	female	4.84	2.24			
AUT_Orig	sample	4.18	3.20	−0.34	0.731	-
	male	4.28	3.44			
	female	4.08	2.97			
Lines_M	sample	0	0.97	−1.00	0.321	-
	male	0.09	−0.09			
	female	−0.09	−0.25			
AUT_M	sample	0.01	0.92	−0.53	0.596	-
	male	0.05	−0.04			
	female	−0.04	−0.04			

Note. * *p* < 0.05; BEN_tot: total psychological well-being; BEN_PS: personal satisfaction; BEN_EC: emotional competence; BEN_CS: coping strategies; CRIq_tot: cognitive reserve total score; CRIq_S: school; CRIq_W: work; CRIq_LA: leisure activities; Flu: fluency; Flex: flexibility; Orig: originality. Lines_M and AUT_M are averaged by z-scores of lines and alternate uses tasks. Significant differences are highlighted in bold.

**Table 2 healthcare-12-00303-t002:** Pearson’s correlations between age and psychological well-being, cognitive reserve, and divergent thinking.

		Age
Lines_Flu	r	−0.36 ***
	*p*-value	<0.001
Lines_Flex	r	−0.33 ***
	*p*-value	<0.001
Lines_Orig	r	−0.30 ***
	*p*-value	<0.001
Lines_M	r	−0.34 ***
	*p*-value	<0.001
AUT_Flu	r	−0.15
	*p*-value	0.098
AUT_Flex	r	−0.17
	*p*-value	0.067
AUT_Orig	r	−0.22 *
	*p*-value	0.017
AUT_M	r	−0.18 *
	*p*-value	0.044
CRIq_S	r	−0.16
	*p*-value	0.081
CRIq_W	r	−0.39 ***
	*p*-value	<0.001
CRIq_LA	r	−0.22 *
	*p*-value	0.018
CRIq_tot	r	−0.34 ***
	*p*-value	<0.001
BEN_PS	r	0.04
	*p*-value	0.625
BEN_EC	r	−0.23 **
	*p*-value	0.010
BEN_CS	r	−0.20 *
	*p*-value	0.027
BEN_tot	r	−0.13
	*p*-value	0.145

Note. * *p* < 0.05; ** *p* < 0.01; *** *p* < 0.001. BEN_tot: total psychological well-being; BEN_PS: personal satisfaction; BEN_EC: emotional competence; BEN_CS: coping strategies; CRIq_tot: cognitive reserve total score; CRIq_S: school; CRIq_W: work; CRIq_LA: leisure activities; Flu: fluency; Flex: flexibility; Orig: originality. Lines_M and AUT_M are averaged by z-scores of lines and alternate uses tasks.

**Table 3 healthcare-12-00303-t003:** Pearson’s correlations between divergent thinking, psychological well-being, and cognitive reserve.

		BEN_tot	BEN_PS	BEN_EC	BEN_CS	CRIq_S	CRIq_W	CRIq_LA	CRIq_tot
CRIq_S	r	0.02	−0.12	0.16	0.00	—			
	*p*	0.797	0.206	0.081	0.978	—			
CRIq_W	r	0.26 **	0.12	0.29 **	0.27 **	0.41 ***	—		
	*p*	0.005	0.195	0.001	0.003	<0.001	—		
CRIq_LA	r	0.13	−0.00	0.28 **	0.06	0.54 ***	0.25 **	—	
	*p*	0.150	0.967	0.002	0.525	<0.001	0.006	—	
CRIq_tot	r	0.20 *	0.02	0.34 ***	0.16	0.78 ***	0.70 ***	0.82 ***	—
	*p*	0.025	0.808	<0.001	0.082	<0.001	<0.001	<0.001	—
Lines_Flu	r	0.11	−0.02	0.16	0.06	0.28 **	0.32 ***	0.30 ***	0.40 ***
	*p*	0.239	0.854	0.075	0.545	0.002	<0.001	<0.001	<0.001
Lines_Flex	r	0.11	−0.03	0.16	0.04	0.25 **	0.35 ***	0.26 **	0.38 ***
	*p*	0.232	0.773	0.072	0.632	0.006	<0.001	0.004	<0.001
Lines_Orig	r	0.12	0.00	0.18 *	0.05	0.12	0.28 **	0.20 *	0.27 **
	*p*	0.190	0.960	0.046	0.622	0.194	0.002	0.029	0.002
Lines_M	r	0.12	−0.01	0.18	0.05	0.23 *	0.33 ***	0.26 **	0.36 ***
	*p*	0.205	0.884	0.055	0.587	0.013	<0.001	0.004	<0.001
AUT_Flu	r	0.06	0.04	0.09	−0.04	0.07	0.10	0.28 **	0.23 *
	*p*	0.540	0.662	0.339	0.626	0.463	0.290	0.002	0.013
AUT_Flex	r	0.09	0.10	0.05	−0.04	0.05	0.08	0.20 *	0.17
	*p*	0.349	0.265	0.589	0.648	0.603	0.391	0.031	0.061
AUT_Orig	r	0.10	0.11	0.11	−0.01	−0.00	0.08	0.21 *	0.16
	*p*	0.287	0.242	0.211	0.899	0.959	0.356	0.021	0.081
AUT_M	r	0.08	0.08	0.09	−0.05	0.05	0.08	0.25 **	0.20 *
	*p*	0.387	0.370	0.317	0.595	0.611	0.376	0.005	0.029
DT_tot	r	0.12	0.04	0.16	0.00	0.17	0.25 **	0.31 ***	0.34 ***
	*p*	0.203	0.673	0.080	0.987	0.070	0.006	<0.001	<0.001

Note. * *p* < 0.05; ** *p* < 0.01; *** *p* < 0.001. BEN_tot: total psychological well-being; BEN_PS: personal satisfaction; BEN_EC: emotional competence; BEN_CS: coping strategies; CRIq_tot: cognitive reserve total score; CRIq_S: school; CRIq_W: work; CRIq_LA: leisure activities; Flu: fluency; Flex: flexibility; Orig: originality. Lines_M and AUT_M are averaged by z-scores of lines and alternate uses tasks.

**Table 4 healthcare-12-00303-t004:** Hierarchical linear regression models considering the effects of cognitive reserve subscales on the different components of psychological well-being (DVs), controlling for age effect.

**DV = BEN_tot**					
**Model**	**R**	**R^2^**	**Adjusted R^2^**	**F**	** *p* **
1	0.13	0.02	0.01	2.15	0.145
2	0.30	0.09	0.06	2.88	0.026 *
**Model comparison**	**R^2^ change**	**F change**	** *p* **		
1–2	0.07	3.08	0.030 *		
**Model coefficients**					
**Model**	**Variables**	**β**	**t**	** *p* **	
1	age	−0.13	−1.47	0.145	
2	age	−0.02	−0.19	0.850	
	CRIq_S	−0.18	−1.57	0.119	
	CRIq_W	0.28	2.71	0.008 **	
	CRIq_LA	0.15	1.43	0.157	
**DV = BEN_PS**					
**Model**	**R**	**R^2^**	**adjusted R^2^**	**F**	** *p* **
1	0.04	0.00	0.00	0.24	0.625
2	0.25	0.06	0.03	1.93	0.111
**Model comparison**	**R^2^ change**	**F change**	** *p* **		
1–2	0.06	2.48	0.064		
**Model coefficients**					
**Model**	**Variables**	**β**	**t**	** *p* **	
1	age	0.04	0.49	0.625	
2	age	1.12	1,22	0.224	
	CRIq_S	−0.25	−2.16	0.033 *	
	CRIq_W	0.24	2.30	0.023 *	
	CRIq_LA	0.09	0.86	0.389	
**DV= BEN_EC**					
**Model**	**R**	**R^2^**	**adjusted R^2^**	**F**	** *p* **
1	0.23	0.06	0.05	6.93	0.010 *
2	0.38	0.14	0.11	4.76	0.001 **
**Model comparison**	**R^2^ change**	**F change**	** *p* **		
1–2	0.09	3.87	0.011		
**Model coefficients**					
**Model**	**Variables**	**β**	**t**	** *p* **	
1	age	−0.23	−2.63	0.010 *	
2	age	−0.11	−1.18	0.240	
	CRIq_S	−0.07	−0.67	0.506	
	CRIq_W	0.21	2.11	0.037 *	
	CRIq_LA	0.24	2.30	0.023 *	
**DV = BEN_CS**					
**Model**	**R**	**R^2^**	**adjusted R^2^**	**F**	** *p* **
1	0.20	0.04	0.03	5.03	0.027 *
2	0.31	0.10	0.07	3.18	0.016 *
**Model comparison**	**R^2^ change**	**F change**	** *p* **		
1–2	0.06	2.49	0.063		
**Model coefficients**					
**Model**	**Variables**	**β**	**t**	** *p* **	
1	age	−0.20	−2.24	0.027 *	
2	age	−0.11	−1.10	0.273	
	CRIq_S	−0.15	−1.38	0.169	
	CRIq_W	0.28	2.68	0.008 **	
	CRIq_LA	0.05	0.46	0.647	

Note. * *p* < 0.05; ** *p* < 0.01. DV = dependent variable; BEN_tot: total psychological well-being; BEN_PS: personal satisfaction; BEN_EC: emotional competence; BEN_CS: coping strategies; CRIq_tot: cognitive reserve total score; CRIq_S: school; CRIq_W: work; CRIq_LA: leisure activities; Flu: fluency; Flex: flexibility; Orig: originality. Lines_M and AUT_M are averaged by lines and alternate uses tasks’ z-scores.

**Table 5 healthcare-12-00303-t005:** Summary of mediation analysis considering cognitive reserve total score as mediator in the relation between the total of divergent thinking skills and the emotional competence subscale of psychological well-being. Indirect and direct effects are summarized below.

	Effect	Estimate	SE	95% C.I. (a)		β	z	Significance (*p*-Value)
				Lower	Upper			
Indirect	DT_tot ⇒ CRIq_tot ⇒ BEN_EC	0.23	0.11	0.02	0.045	0.07	2.16	0.031 *
	age ⇒ CRIq_tot ⇒ BEN_EC	−0.06	0.03	−0.11	−0.00	−0.07	−2.08	0.038 *
Component	age ⇒ CRIq_tot	−0.80	0.27	−1.32	−0.26	−0.26	−2.97	<0.001 ***
	CRIq_tot ⇒ BEN_EC	0.07	0.02	0.03	0.11	0.29	3.14	0.002 **
	DT_tot ⇒ CRIq_tot	3.31	0.98	1.40	5.25	0.25	3.37	<0.001 ***
Direct	age ⇒ BEN_EC	−0.10	0.07	−0.24	0.04	−0.13	−1.41	0.158
	DT_tot ⇒ BEN_EC	0.07	0.32	−0.57	0.70	0.02	0.21	0.831
Total	age ⇒ BEN_EC	−0.16	0.08	−0.31	−0.01	−0.21	−2.04	0.041 *
	DT_tot ⇒ BEN_EC	0.30	0.31	−0.30	0.92	0.10	0.97	0.331

Note. * *p* < 0.05; ** *p* < 0.01; *** *p* < 0.001. Confidence intervals computed with parametric bootstrap method. Betas are completely standardized effect sizes.

## Data Availability

The data presented in this study are available on request from the corresponding author.

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
