# Peer review of "Can Creativity and Cognitive Reserve Predict Psychological Well-Being in Older Adults? The Role of Divergent Thinking in Healthy Aging"

_healthcare, 2024, doi:10.3390/healthcare12030303_

Round 1
Reviewer 1 Report
Comments and Suggestions for Authors
1. Please define healthy older adults
2. Please provide Full-name of BEN-SSC
3. How data collection was conducted
4. The elderly population (in developed countries) is defined as people aged 65 and over, please justify why 60 years and older has been considered as elderly population
5. Please describe study design and used sampling technique to obtain samples
6. Please provide the internal consistency of used questionnaires for the current study
Reviewer 2 Report
Comments and Suggestions for Authors
Line 52: The sentence “However, it has also been associated to improved levels of …” needs working on. Maybe it should be “However, aging has also been associated with improved levels of …”
Line 67: The sentence “overcome the only emotional sphere …” is unclear.
Line 179-180: Consider replacing the word ‘disease’ with ‘condition’ or ‘disorder’ instead of having ‘psychiatric disease.’
Line 202: For the BEN-SSC are scores meant to be added to obtain a sum total? I am not familiar with the BEN-SSC so am checking.
Line 206: Should ‘Cri-q’ be written in upper case.
Line 214: For the CRI-Q are scores meant to be added to obtain a sum total?
Line 282: There seems to be an error in formatting for the first row of Table 3 (BEN PS column). This cell entry has a dash in it. Further, I could not understand what the two rows mean for the p values. For example, for the BeEN_tot and CRIq_S cell, I see that .7 is a probability, but what is 97?
Line 289: Instead of “According to our hypothesis …” write “Consistent with our hypothesis …”
Line 318: The sentence “only one and marginal correlation can be observed between …” does not read right. Should it be “only one marginal correlation …”? Further, consider replacing ‘can be’ with ‘was’.
Line 320: Consider rewriting “No other correlation was found between DT skills and PWB” as “No other correlation between DT skills and PWB were found to be statistically significant.”
Line 321&322: Consider rewriting “However, considering the results found in the previous paragraphs, i.e., DT was positively correlated with CR and CR was, in turn, positively correlated with well-being, we” as “However, considering the results described in the previous paragraphs (i.e., DT was positively correlated with CR and CR was, in turn, positively correlated with well-being), we …”
Line 336: Consider rewriting: “We found that BEN_EC was not directly predicted nor by age nor by DT_tot” as “We found that BEN_EC was neither directly predicted by age nor DT_tot.”
Line 450: Delete the words “can”.
Line 465: Consider having a ‘Strengths and Limitations’ section where you can list both strengths and limitations.
Line 467: Consider replacing “as concerns” with “for”.
Consider shortening the discussion to make it more focussed. Also when referring to correlations, mention if they were statistically significant.
Comments on the Quality of English LanguageEnglish was mostly fine
Reviewer 3 Report
Comments and Suggestions for Authors
The authors conducted a very relevant study of the role of divergent thinking and cognitive reserve in variations of psychological well-being in old age. It should be noted that the authors have analyzed the previous research on the subject very qualitatively and in detail. The methodological part is represented by specific methods, the description of which is quite informative from the point of view of the instrumental equipment of the study. The Results section is presented with a thorough analysis of all level indicators, correlations and regressions. The section "Discussion" is presented by comparing the data obtained with those available in science, as well as a qualitative interpretation of the results.
At the same time, the article would have been presented more qualitatively if the authors had paid attention to the studies of "positive aging" in the review part. The second point is that the reference to the Carroll Ryff approach may not be entirely correct, since it causes some embarrassment that the indicators of the scale used do not match the Ryff scales (Psychological well-being consists of self-acceptance, positive relationships with others, autonomy, environmental mastery, a feeling of purpose and meaning in life, and personal growth and development). Table 1 shows the average results for the sample, but analyzes the sex differences. It would be appropriate to include the average and standard deviations of the indicators of men and women. It is also necessary to format the entire text (some parts have different line spacing).
I wish success to the authors in publishing the article in the journal.
